# Beyond Majority Voting:
# Self-Reflective Test-Time Reinforcement Learning for LLM Reasoning

**Sitong Wu**[1]  **Haoru Tan**[2]  **Xichen Zhang**[3]  **Bin Xia**[1]  **Shaofeng Zhang**[4]  **Xiaojuan Qi**[2]  **Bei Yu**[1]  **Jiaya Jia**[✉ 1 3]

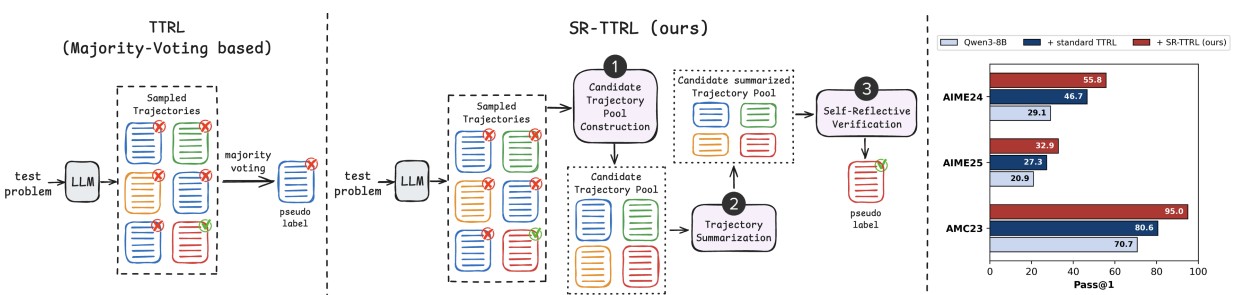

Figure 1. Comparison of SR-TTRL and standard TTRL (Zuo et al., 2025) (based on majority-voting). *(left & middle)* Design philosophy of pseudo-labeling mechanism. The trajectories with different final answers are distinguished by color. *(right)* Performance comparison.

## Abstract

The core challenge of *Test-Time Reinforcement Learning* (TTRL) lies in estimating rewards without access to ground-truth supervision. Existing TTRL methods predominantly rely on majority voting to generate pseudo-labels, under the assumption that the most frequent answer among sampled trajectories is correct. However, we observe that this assumption frequently breaks down in complex reasoning tasks, where correct solutions often constitute a logical minority. As a result, rare yet correct trajectories are systematically undervalued by majority-voting-based approaches. To address this limitation, we propose **Self-Reflective Test-Time Reinforcement Learning (SR-TTRL)**, a novel framework that leverages self-reflective verification to produce high-fidelity pseudo-labels. Specifically, given multiple sampled trajectories for a prob-
lem, SR-TTRL first groups trajectories according to their final answers and selects one representative from each group to form a candidate pool. Each candidate trajectory is then summarized to preserve its core reasoning steps while reducing verbosity. Finally, the model performs self-reflection over the candidate pool, critically evaluating and selecting the most plausible trajectory as the pseudo-label. Empirically, SR-TTRL achieves substantially higher pseudo-label fidelity and sample efficiency than prior majority-voting-based TTRL methods. Extensive experiments across diverse benchmarks and model families demonstrate that SR-TTRL consistently outperforms majority-voting baselines and significantly improves generalization to novel problems. For example, SR-TTRL improves the Pass@1 accuracy of Qwen3-8B on AIME24 from 29.1 to 55.8 (a gain of +26.7), exceeding standard TTRL by an additional +9.1. The code will be released at: https://github.com/JIA-Lab-research/SR-TTRL.

[1]Department of Computer Science and Engineering, the Chinese University of Hong Kong [2]Department of Electrical and Computer Engineering, the University of Hong Kong [3]Department of Computer Science and Engineering, the Hong Kong University of Science and Technology [4]Shanghai Jiao Tong University. Correspondence to: Jiaya Jia <leojia@cse.cuhk.hk>.

*Proceedings of the 43rd International Conference on Machine Learning*, Seoul, South Korea. PMLR 306, 2026. Copyright 2026 by the author(s).

## 1. Introduction

Recent advances in Large Language Models (LLMs) have revealed the immense potential of Reinforcement Learning (RL) for enhancing complex reasoning capabilities. Models

like OpenAI o1 (OpenAI, 2024) and DeepSeek-R1 (Guo et al., 2025) have demonstrated remarkable performance on challenging reasoning tasks through RL training (Guo et al., 2025; Liu et al., 2025; Yu et al., 2025b; Chu et al., 2025b; Shao et al., 2024a; Yu et al., 2025a; Hu et al., 2025a). However, these approaches predominantly rely on expensive human-annotated data or explicitly verifiable ground-truth labels, creating a fundamental generalization bottleneck, especially on challenging or unseen problems. This limitation has motivated the emerging paradigm of Test-Time Reinforcement Learning (TTRL), which enables LLMs to self-evolve through experience only on test data.

The core challenge in TTRL lies in reward estimation without access to the ground-truth answer. To address this, existing TTRL approaches leverage majority voting to generate pseudo-labels from multiple model responses (Zuo et al., 2025). Despite its intuitive appeal, majority voting constitutes a fundamental limitation: it operates on the assumption of "statistical consensus", namely that the correct solution will dominate the sampling distribution, that is, the most frequent answer is the correct one. However, this assumption often breaks down on complex reasoning tasks where models struggle with uncertainty and exploration. We argue that for challenging problems, the truth is not necessarily held by the majority, but is frequently embodied by the logical minority (a correct yet underrepresented reasoning trajectory, even if it appears less frequently among sampled responses).

Our empirical evidence in Sec. 3 strongly supports this view. As shown in Figure 2(a-b), a significant gap exists between the model's potential (measured by Pass@$k$) and its statistical consensus (measured by Maj@$k$). For example, on the challenging AIME25 benchmark, Qwen3-8B achieves a Pass@64 of 63.3%, yet its Maj@64 is only 30.0%. This 33.3 percent gap implies that for approximately 53% of problems within the model's capability (*i.e.,* those for which at least one correct response is generated), the correct answer fails to form a statistical majority and is consequently discarded by majority-voting based TTRL. These findings reveal a critical flaw in the current TTRL framework (Zuo et al., 2025): its reward mechanism, which relies on majority-voting pseudo-labels, is fundamentally constrained by the frequency of correct answers rather than the intrinsic correctness of the reasoning. As a result, high-quality responses from the logical minority are systematically suppressed, imposing an artificial ceiling on self-improvement.

This observation leads us to a key insight: for complex reasoning tasks, reliable pseudo-labeling demands validation through rigorous analysis of solution trajectories. The correct answer is not necessarily the most common one, but rather the one supported by the most sound, coherent, and deductively valid reasoning process. This insight raises a fundamental question:

> *Can we replace statistical consensus with logical consensus, where the model itself evaluates the quality of different reasoning paths?*

In this paper, we propose Self-Reflective Test-Time Reinforcement Learning (**SR-TTRL**), a novel framework that constructs high-fidelity pseudo-labels using a logical self-reflective mechanism instead of statistical majority voting. The core idea is to leverage the model's intrinsic reasoning and reflection capability to analyze, critique, and select the most plausible one among diverse candidate responses. Specifically, our self-reflective pseudo-labeling mechanism consists of three key steps: (1) **Candidate Trajectory Pool Construction.** Given a problem, we first sample multiple reasoning trajectories from the current policy model. We then cluster the sampled trajectories by their final answers and randomly select one representative trajectory from each cluster to form a compact yet diverse candidate pool. (2) **Trajectory Summarization.** For each trajectory in the candidate pool, we prompt the model to condense it into a concise yet logically complete summary. The summarization preserves critical reasoning steps and intermediate conclusions while significantly reducing redundant tokens to fit within the model's context window in the next step. (3) **Self-Reflective Verification.** We feed all the summarized trajectories back into the model itself under a structured reflection prompt, instructing it to compare, critique, and select the single most reliable reasoning path based on logical correctness and reasoning quality. The final answer of the selected trajectory is then adopted as the pseudo-label, which further serves as the target for reward estimation.

Our SR-TTRL offers three key advantages over the standard majority-voting based TTRL: (1) **Higher Pseudo-Label Fidelity.** SR-TTRL produces more accurate pseudo-labels than majority voting (Figure 4(a)), which stems from its reliance on the more robust logical self-reflection over reasoning trajectories rather than unstable statistical frequency. This principled mechanism enables it to recover correct solutions from the logical minority, precisely the cases where majority voting fails. (2) **Performance Superiority.** Extensive experiments demonstrate the consistent superiority of SR-TTRL across diverse benchmarks and model families (Table 1). For example, SR-TTRL improves Qwen3-8B's accuracy from 29.1 to 55.8 (with a +26.7 gain) on the challenging AIME24 benchmark, surpassing the majority-voting baseline by +9.1. These improvements arise from the higher-fidelity pseudo-labels, which further contribute to more accurate reward signals and enable more effective and stable policy updates. (3) **Higher Efficiency.** Our SR-TTRL achieves comparable performance with only 8 samples (Figure 4(b)). This 8× reduction in sampling cost dramatically lowers test-time compute overhead and latency, making continuous online self-improvement more practical.

## 2. Related Work

**Reinforcement Fine-Tuning for LLM Reasoning.** Extensive empirical evidence (OpenAI, 2024; Guo et al., 2025; Yang et al., 2024) has demonstrated that Reinforcement Fine-Tuning (RFT) (Schulman et al., 2017; Shao et al., 2024a; Yu et al., 2025a; Hu et al., 2025a) is a critical phase in enhancing the reasoning capabilities of Large Language Models (LLMs). Unlike Supervised Fine-Tuning (SFT) (Chu et al., 2025a), RFT optimizes for the maximization of verifiable rewards, which can be as straightforward as binary correctness signals (Guo et al., 2025). Numerous studies have established that RFT is pivotal for strengthening a model's generalizability (Chu et al., 2025a) and catalyzing its emergent self-reflection capabilities. (Gandhi et al., 2025; Guo et al., 2025). Current research on Reinforcement Fine-Tuning primarily focuses on three dimensions: (1) enhancing algorithmic efficiency and robustness (Liu et al., 2025; Yu et al., 2025b; Chu et al., 2025b; He et al., 2025), as exemplified by DAPO (Yu et al., 2025a) and REINFORCE++ citereinforce++; (2) curating core datasets (Li et al., 2025; Luo et al., 2025; Wang et al., 2025b) to optimize training efficiency and performance; and (3) designing advanced reward mechanisms (Wang et al., 2024; Zheng et al., 2023), such as fine-grained the process-level (Uesato et al., 2022; Li et al., 2023) and even token-levels (Cui et al., 2025; Lyu et al., 2025; Lee et al., 2024). .

**Test-Time Training (TTT).** This technique has a long developmental history. Its origins can be traced back to early efforts in image classification, in which researchers performed test-time adaptation by updating the statistics of normalization layers (Wang et al., 2021; Li et al., 2016) or by optimizing lightweight self-supervised auxiliary objectives during inference (Sun et al., 2020). Recent findings (Akyürek et al., 2025; Hübotter et al., 2025; Hardt & Sun, 2024; Hu et al., 2025b) highlight TTT as a powerful approach to mitigate distribution shifts in LLMs. The underlying optimization targets are typically elegant and simple, such as self-supervised perplexity-based adaptation (Hu et al., 2025b). TTRL (Zuo et al., 2025) formulates test-time inference as a local RL problem, using majority voting over multiple rollouts to derive pseudo-labels as reward signals, enabling self-evolution on unlabeled data. Subsequent research has expanded upon these foundations, with emerging work exploring Test-Time Training in Agent-based environments (Acikgoz et al., 2025; Xue et al., 2025) and seeking to optimize the training efficiency of TTRL-style frameworks (Yuksekgonul et al., 2026; Wang et al., 2025a).

## 3. The Limits of Statistical Consensus

In this section, we examine the reliability of majority voting for pseudo-label generation in test-time reinforcement learning. Through pilot experiments, we identify two fundamental bottlenecks, namely a performance ceiling and a scalability paradox, that limit pseudo-label quality under majority-voting-based mechanisms. These findings motivate our framework, which shifts the focus from frequency-based selection to qualitative logical verification.

### 3.1. The Phenomenon of Logical Minority

Our primary observation is that in complex reasoning tasks, the correct solution often fails to achieve a statistical majority, instead becoming a "logical minority". As illustrated in Figure 2(a), we compare Pass@64 (reflecting the model's potential) with Maj@64 (measuring the accuracy of statistical consensus) for the Qwen3-8B model. When sampling 64 trajectories per problem, a significant gap between Pass@64 and Maj@64 exists across all benchmarks. For example, on AIME24, Pass@64 reaches 80.0, yet Maj@64 is only 43.3, resulting in a staggering 36.7 absolute gap. Similarly, on AIME25, the gap remains as high as 33.3 (63.3 vs. 30.0). More importantly, among the problems that the model has the capacity to solve, 45.9% of the problems in AIME24 and 52.6% in AIME25 fall into the category where the correct answer is *not* the majority. This implies that for approximately half of the solvable problems, the ground-truth resides in the logical minority, where it is statistically overshadowed by incorrect alternatives.

This phenomenon indicates that while the model has the capability to solve the problem (it generates the correct answer at least once), the purely statistics-based Majority Voting mechanism is insufficient to identify the truth, as it lacks the capacity to scrutinize the underlying logic beyond simple frequency counts.

**Reason Analysis.** This phenomenon can be largely attributed to the evolving distribution of sampled trajectories as task complexity increases. As illustrated in Figure 2(c-e), the answer frequency typically transitions from a *unimodal* state to a *multimodal* or even *skewed* distribution, which accounts for the failure of statistical consensus:

- *Unimodal (Dominant Truth):* On simpler problems, the reasoning paths are relatively direct, allowing the correct answer to act as a strong attractor. In such cases, the distribution is unimodal, and statistical consensus serves as a reliable proxy for the ground-truth, as Figure 2(c).
- *Multimodal (Increased Uncertainty):* As the reasoning depth increases, the model's uncertainty may accumulate, causing the probability mass to disperse across several competing incorrect answers. This resulting multimodal distribution leads to a fragmented consensus where no single answer holds a clear majority, as Figure 2(d).
- *Skewed (Logical Traps):* For some problems with subtle logical traps that mislead the model, the answer distribution often becomes skewed, as Figure 2(e). Unlike ran-

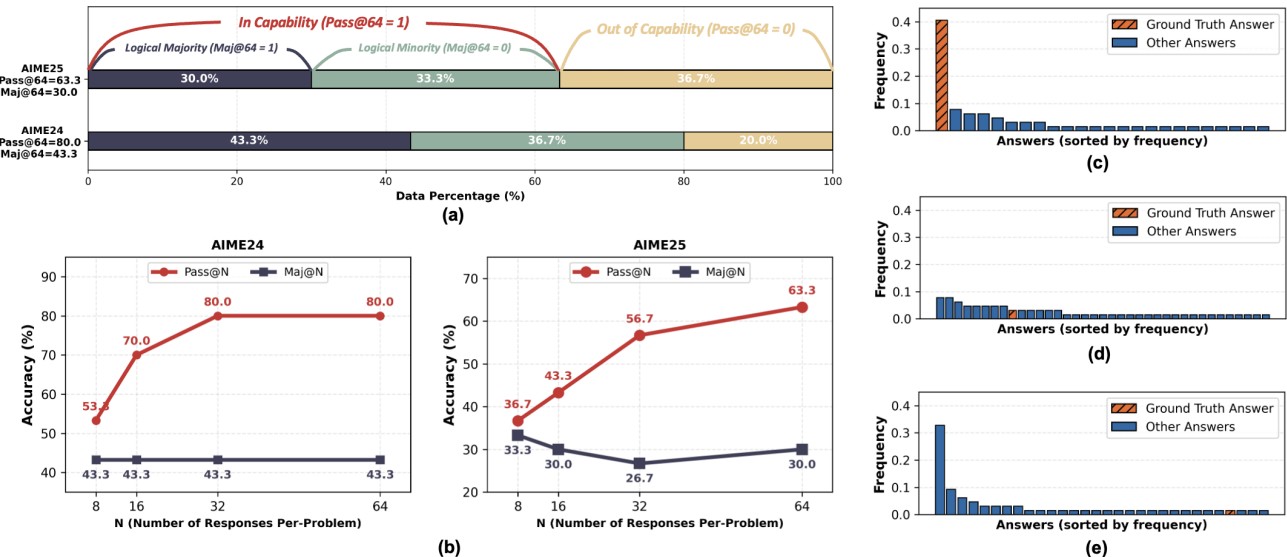

*Figure 2.* Analysis of the unreliability of the majority voting pseudo-labeling mechanism. **(a)** Illustration of the "logical minority" phenomenon. Although the model can solve 63.3% test problems in AIME25 within the red bracket (Pass@64=1), the majority voting only identifies 30% instances in the dark blue segment. The significant proportion (33.3%) of the green segment quantifies the gap, solvable problems where the correct answer is a statistical minority. **(b)** Scaling behavior of Pass@$N$ and Maj@$N$ on AIME24 and AIME25 benchmarks as $N$ increases. While Pass@$N$ grows significantly with $N$, Maj@$N$ stagnates or fluctuates, revealing the poor scalability of majority voting for improving pseudo-label quality. **(c-e)** Three typical answer frequency distributions. *(c) Unimodal:* Correct answer dominates. *(d) Multimodal:* Probability mass is scattered due to high uncertainty. *(e) Skewed:* A logical trap creates a false consensus on an incorrect answer, surpassing the correct one. All the results are derived from the Qwen3-8B model.

dom errors that tend to disperse across the answer space, these systematic errors cluster around specific incorrect answers, creating a false consensus. In such scenarios, an erroneous trajectory can statistically outrank the correct one, relegating the truth to a logical minority.

These observations underscore that as we move toward harder reasoning frontiers, the "wisdom of the crowd" becomes increasingly deceptive. High-quality pseudo-labels can no longer be derived from quantitative counting; instead, they require a qualitative verification mechanism capable of recognizing the logical rigor of a minority solution.

### 3.2. The Poor Scalability of Statistical Consensus

Our empirical analysis reveals a striking scalability paradox of majority voting: increasing $N$ dramatically boosts Pass@N (model capability) but leaves Maj@$N$ nearly unchanged. As illustrated in Figure 2(b), scaling $N$ from 8 to 64 boosts the Pass@$N$ from 53.3 to 80.0 on AIME24, indicating that the model is indeed capable of uncovering correct solutions given more attempts. In stark contrast, the Maj@N remains completely flat at 43.3. A similar trend is observed on AIME25, where the potential increases by 26.6 (from 36.7 to 63.3) while the consensus accuracy plateaus at 30.0. These results confirm that majority voting has reached a scaling ceiling in complex reasoning tasks for test-time pseudo-labeling.

This stagnation occurs because majority voting is a quantity-driven mechanism that cannot resolve quality gaps. In multi-modal or skewed distributions (Sec. 3.1), additional samples merely amplify existing noise without increasing the relative frequency of correct answers. Worse, in skewed regimes, more samples strengthen the dominance of logical traps, actively degrading pseudo-label quality.

### 3.3. Implications for TTRL

These two phenomena impose a dual bottleneck on conventional majority-voting-based TTRL:

- *Performance Ceiling*: The reward signal quality is fundamentally capped at Maj@$N$, far below the model's true capability (Pass@$N$). On AIME24, this ceiling is 43.3 despite a solvable rate of 80.0, preventing TTRL from learning from correct trajectories in the logical minority.
- *Scalability Failure*: Increasing the number of samples $N$ fails to improve pseudo-label accuracy, as Maj@$N$ stagnates even when $N$ grows eightfold (*e.g.,* from 8 to 64). This renders large-scale sampling computationally wasteful, as additional trajectories only amplify statistical noise without enhancing consensus quality.

Together, these bottlenecks create a paradox: the more the model struggles with a problem, the less likely it is to learn from its own correct attempts. To break this cycle, we need a

quantitative to qualitative verification mechanism, that is, a shift from statistical counting to principled logical analysis. This insight directly motivates our Self-Reflective TTRL framework.

# 4. Method

In this section, we present the technical details of Self-Reflective Test-Time Reinforcement Learning (SR-TTRL). Building on the observation that logical truth often resides in the minority of sampled trajectories (as analyzed in Sec. 3), SR-TTRL shifts the pseudo-labeling mechanism from statistical frequency to self-reflective logical verification.

## 4.1. Framework Overview

SR-TTRL operates as an iterative online learning loop over unlabeled test data. For each problem $x$, the framework executes three sequential phases (see Figure 3):

- **Exploration:** We sample $N$ diverse reasoning trajectories from the old policy model $\pi_{\theta_{\text{old}}}$:

$$\mathcal{Y} = \{y_i \mid y_i \sim \pi_{\theta_{\text{old}}}(\cdot \mid x)\}_{i=1}^{N} \quad (1)$$

- **Self-Reflective Pseudo-Labeling:** Instead of majority voting, we prompt the model to perform structured self-reflection over the sampled trajectories, selecting the most logically sound trajectory $\hat{y}$ from $\mathcal{Y}$ and treating its final answer $\hat{p}$ as the pseudo-label (detailed in Sec. 4.2):

$$\hat{y}, \hat{p} = \mathcal{F}(x, \mathcal{Y}) \quad (2)$$

- **Policy Optimization:** We qualify each trajectory in $\mathcal{Y}$ against the reflective pseudo-label $\hat{p}$ to compute rewards, and then update the policy model $\pi_{\theta}$ via policy optimization algorithms (Shao et al., 2024b) by maximizing the expected reward (detailed in Sec. 4.3):

This iterative training cycle (exploration, self-reflective pseudo-labeling, and policy optimization) forms a virtuous cycle of self-improvement. As the model's policy becomes more capable, the accuracy of both its reasoning trajectories and its self-reflection ability increases, which in turn provides even higher-fidelity pseudo-labels for subsequent training steps.

By replacing the statistical consensus with a self-reflective mechanism, SR-TTRL effectively utilizes the model's latent reasoning potential (Pass@$N$) to guide its own improvement, even when the correct trajectory is not the most frequent.

## 4.2. Self-Reflective Pseudo-Labeling

The self-reflective pseudo-labeling mechanism $\mathcal{F}$ consists of three sequential steps designed to identify the most logically sound reasoning paths.

**Step 1: Candidate Trajectory Pool Construction**

Given $N$ sampled trajectories $\mathcal{Y} = \{y_1, y_2, \ldots, y_N\}$ for problem $x$, we first group them by their final answers. Let $\mathcal{A} = \{a_1, a_2, \ldots, a_G\}$ denote the set of $G$ distinct final answers extracted via the answer extractor function $\mathcal{E}(\cdot)$. The trajectories are partitioned into $G$ disjoint clusters:

$$\mathcal{C}_g = \{y_i \in \mathcal{Y} \mid \mathcal{E}(y_i) = a_g\}, \quad g = 1, \ldots, G. \quad (3)$$

To eliminate redundancy and ensure computational efficiency for subsequent steps, we randomly select one representative trajectory $\tilde{y}_g$ from each cluster $\mathcal{C}_g$. The resulting candidate pool is:

$$\mathcal{P} = \{\tilde{y}_1, \tilde{y}_2, \ldots, \tilde{y}_G\}. \quad (4)$$

This construction guarantees that $\mathcal{P}$ contains exactly one trajectory per unique answer, preserving solution diversity while minimizing unnecessary repetition.

**Step 2: Trajectory Summarization**

Directly feeding multiple full-length trajectories from the candidate pool $\mathcal{P}$ into the model for comparative reflection is often impractical and ineffective. As reasoning trajectories for complex tasks can span thousands of tokens, presenting several such responses simultaneously may exceed the model's effective context window and introduce excessive stylistic noise that distracts from logical reflection.

To address this, we introduce a summarization operation $\mathcal{S}$ to condense each representative trajectory into a logical summary:

$$s_g = \mathcal{S}(\tilde{y}_g, P_{\text{sum}}), \quad g = 1, \ldots, G, \quad (5)$$

where $\mathcal{S}$ is implemented via prompt engineering without additional training. $P_{\text{sum}}$ is a specialized prompt designed to extract the core reasoning skeleton.

To address this, we introduce a summarization operation $\mathcal{S}$ to condense each representative trajectory into a logical summary $\check{y}_g$:

$$\check{y}_g = \mathcal{S}(\tilde{y}_g, P_{\text{sum}}), \quad g = 1, \ldots, G, \quad (6)$$

where $\mathcal{S}$ is implemented via prompt engineering without additional training. $P_{\text{sum}}$ is a specialized prompt designed to extract the core reasoning skeleton (see Appendix A.1 for details). These summarized trajectories collectively constitute a candidate summarized trajectory pool:

$$\check{\mathcal{P}} = \{\check{y}_1, \check{y}_2, \ldots, \check{y}_G\}. \quad (7)$$

This summarization process is instructed to retain critical mathematical derivations, key intermediate steps, decision points, and final answer, while systematically filtering out

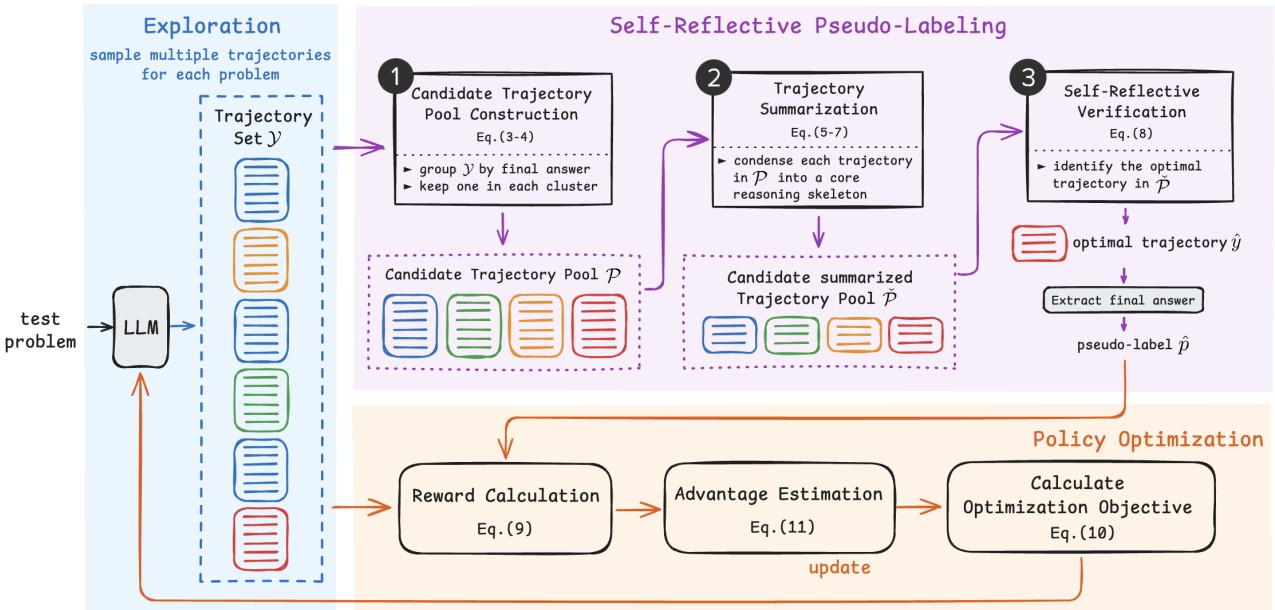

*Figure 3.* Framework of our Self-Reflective Test-Time Reinforcement Learning (SR-TTRL). It consists of three sequential phases: *(1) Exploration:* We sample multiple trajectories for each problem. The trajectories with different final answers are distinguished by color in the figure. *(2) Self-Reflective Pseudo-Labeling:* We find the most likely correct trajectory via a self-reflection process, and the pseudo-label is defined as the final answer of such an optimal trajectory. This phase is the core innovation of our SR-TTRL, which we detail in Sec. 4.2. *(3) Policy Optimization:* Given the pseudo-label derived in the last phase, we calculate the reward and update the policy model via group-relative policy optimization algorithm.

redundant information, such as auxiliary explanations, invalid exploration paths, and erroneous steps that were subsequently self-corrected within the original trajectory. By reducing the total token count of the candidates while preserving their deductive integrity, this step ensures that the subsequent reflective selection can be conducted within a concise and focused reasoning trajectory.

**Step 3: Self-Reflective Verification**

This step leverages the policy model's inherent capabilities to identify the optimal trajectory from the candidate summary pool $\check{\mathcal{P}}$ via a self-reflection process.

Specifically, we construct a self-reflection prompt $P_{\text{ref}}$ (refer to Appendix A.2 for details) that presents the original problem $x$ alongside all candidate summarized trajectories in $\check{\mathcal{P}}$. The policy model $\pi_\theta$ is then instructed to perform a cross-comparison, examining each $\check{y}_g$ for logical correctness. This process yields a selection decision $\hat{k}$:

$$\hat{k} = \text{Reflect}(\pi_\theta, x, \check{\mathcal{P}}, P_{\text{ref}}) \quad \hat{k} \in \{1, \ldots, G\}, \quad (8)$$

where $\hat{k}$ represents the index of the summary deemed most reliable by the model. The final answer associated with the selected candidate, $a_{\hat{k}}$, is subsequently adopted as the pseudo-label $\hat{p}$ for the problem $x$.

By evaluating trajectories qualitatively rather than quantitatively, this step enables SR-TTRL to pinpoint the logical

minority, *i.e.,* correct reasoning paths that are underrepresented in the sampling distribution.

**4.3. Policy Optimization with Pseudo-Labels**

With the pseudo-label $\hat{p}$ derived from self-reflective selection, we compute rewards for all sampled trajectories in $\mathcal{Y}$ and update the policy via on-policy reinforcement learning.

Specifically, for each trajectory $y_i$ in the original sampled set $\mathcal{Y} = \{y_1, y_2, \ldots, y_N\}$, we first assign a binary reward $r_i$ based on the consistency of its final answer $\mathcal{E}(y_i)$ with the reflective pseudo-label $\hat{p}$: $r_i = \mathbb{I}\left[\mathcal{E}(y_i) = \hat{p}\right]$, where $\mathbb{I}[\cdot]$ is the indicator function and $\mathcal{E}(\cdot)$ extracts the final answer. This allows the model to receive positive reinforcement even for trajectories that belong to a logical minority, provided they are verified as correct by the self-reflection process.

Then, the policy model $\pi_\theta$ is optimized by the objective function as follows:

$$\mathcal{J} = \sum_{i=1}^{N} \frac{1}{N|y_i|} \sum_{t=1}^{|y_i|} A_i \min\left(\rho_{i,t}(\theta), \text{clip}\left(\rho_{i,t}(\theta), 1 - \epsilon, 1 + \epsilon\right)\right), \quad (9)$$

where $|y_i|$ is the length of trajectory $y_i$. The importance sampling ratio $\rho_{i,t}(\theta) = \frac{\pi_\theta(y_{i,t}|y_{i,<t},x)}{\pi_{\theta_{\text{old}}}(y_{i,t}|y_{i,<t},x)}$ is designed to correct the distributional shift between old policy $\pi_{\theta_{\text{old}}}$ that sampled trajectories and current policy $\pi_\theta$ being optimized. The clip

*Table 1.* Comparison of TTRL (Zuo et al., 2025) and our SR-TTRL on various models and complex reasoning benchmarks.

| Model | AIME24 | AIME25 | AMC23 | MATH-500 | Avg. |
|---|---|---|---|---|---|
| **Qwen3-1.7B** | 13.4 | 9.8 | 49.3 | 73.0 | 36.4 |
| + TTRL | 19.8 (+6.4) | 15.6 (+5.8) | 50.2 (+0.9) | 84.9 (+11.9) | 42.6 (+6.2) |
| + SR-TTRL (ours) | **26.3 (+12.9)** | **20.4 (+10.6)** | **66.5 (+17.2)** | **89.3 (+16.3)** | **50.6 (+14.2)** |
| Δ (ours - TTRL) | **+6.5** | **+4.8** | **+16.3** | **+4.4** | **+8.0** |
| **Qwen3-4B** | 25.0 | 19.1 | 68.3 | 84.8 | 49.3 |
| + TTRL | 35.8 (+10.8) | 27.0 (+7.9) | 78.4 (+10.1) | 85.5 (+0.7) | 56.7 (+7.4) |
| + SR-TTRL (ours) | **48.5 (+23.5)** | **32.0 (+12.9)** | **92.0 (+23.7)** | **90.4 (+5.6)** | **65.7 (+16.4)** |
| Δ (ours - TTRL) | **+12.7** | **+5.0** | **+13.6** | **+4.9** | **+9.0** |
| **Qwen3-8B** | 29.1 | 20.9 | 70.7 | 87.4 | 52.0 |
| + TTRL | 46.7 (+17.6) | 27.3 (+6.4) | 80.6 (+9.9) | 89.3 (+1.9) | 61.0 (+9.0) |
| + SR-TTRL (ours) | **55.8 (+26.7)** | **32.9 (+12.0)** | **95.0 (+24.3)** | **91.9 (+4.5)** | **68.9 (+16.9)** |
| Δ (ours - TTRL) | **+9.1** | **+5.6** | **+14.4** | **+2.6** | **+7.9** |
| **LLaMA3.2-3B-Instruct** | 3.8 | 0.3 | 20.5 | 43.9 | 17.1 |
| + TTRL | 13.3 (+9.5) | 0.3 (+0.0) | 31.3 (+10.8) | 61.6 (+17.7) | 26.6 (+9.5) |
| + SR-TTRL (ours) | **20.0 (+16.2)** | **3.3 (+3.0)** | **47.5 (+27.0)** | **67.5 (+23.6)** | **34.9 (+17.8)** |
| Δ (ours - TTRL) | **+6.7** | **+3.0** | **+16.2** | **+5.9** | **+8.3** |
| **LLaMA3.1-8B-Instruct** | 4.6 | 1.3 | 23.3 | 48.6 | 19.4 |
| + TTRL | 10.0 (+5.4) | 3.3 (+2.0) | 32.3 (+9.0) | 63.7 (+15.1) | 27.3 (+7.9) |
| + SR-TTRL (ours) | **24.2 (+19.6)** | **6.7 (+5.4)** | **51.5 (+28.2)** | **72.8 (+24.2)** | **38.5 (+19.1)** |
| Δ (ours - TTRL) | **+14.2** | **+3.4** | **+19.2** | **+8.1** | **+11.2** |

operation $\text{clip}(\cdot, 1 - \epsilon, 1 + \epsilon)$ constrains the magnitude of policy update for training stability, with $\epsilon$ as the clipping threshold. Following GRPO (Shao et al., 2024b), the advantage term $A_i$ is computed relative to the group of trajectories for the same problem: $A_i = \frac{r_i - \text{mean}\left(\{r_1, r_2, ..., r_N\}\right)}{\text{std}\left(\{r_1, r_2, ..., r_N\}\right)}$.

# 5. Experiments

The experiment settings and implementation details are provided in Sec. 5.1. Sec. 5.2 analyzes the main results, and Sec. 5.4 provides more discussions. Key designs are ablated in Sec. 5.3.

## 5.1. Experiment Settings

**Baseline Models.** To ensure comprehensive evaluation, we conduct experiments across different scale variants of two prominent open-source model families: Qwen and LLaMA. For the Qwen family, we select the latest Qwen3 series (Yang et al., 2025), including Qwen3-1.7B, Qwen3-4B, and Qwen3-8B. For the LLaMA family (Dubey et al., 2024), we use the most recent instruction-tuned models: LLaMA3.2-3B-Instruct and LLaMA3.1-8B-Instruct.

**Training.** We adopt the AdamW optimizer with a weight decay of 0.01 and a peak learning rate of $5 \times 10^{-7}$, warmed up over the first 10 steps, followed by a cosine decay schedule. By default, for both standard TTRL and our SR-TTRL, we sample $N = 32$ reasoning trajectories per problem, with a maximum generation length of 8,192 tokens. The number of episodes is set according to task difficulty: 10 for MATH-500, 30 for AMC23, and 80 for AIME24 and AIME25.

Each training batch consists of a single problem instance (batch size = 1). All experiments are conducted on 8 × A100 GPUs.

**Evaluation.** We evaluate our method on four widely used mathematical reasoning benchmarks: AIME24 (AIME, 2024), AIME25 (AIME, 2025), AMC23 (AMC, 2023), and MATH-500 (Hendrycks et al., 2021). Among these, AIME24 and AIME25 are particularly challenging, featuring competition-level problems that require deep multi-step reasoning. In Table 1, the Qwen3 models are evaluated in non-thinking mode. We report Pass@1 accuracy, computed as the average success rate over 8 independent samples per problem for all models. Sampling configurations follow model-specific best practices: for LLaMA models, we use temperature = 0.6, top-$p$ = 0.95, and top-$k$ = −1; for Qwen3 models, we adopt the official recommended settings with temperature = 0.7, top-$p$ = 0.8, and top-$k$ = 20. All generations are capped at a maximum length of 8,192 tokens.

## 5.2. Main Results

**Significant Self-Improvement Effect.** Our SR-TTRL demonstrates remarkable capability in enhancing baseline models, validating the necessity of test-time learning for unlocking reasoning potential on unseen problems. As shown in Table 1, shifting from static inference to test-time adaptation yields dramatic gains. For instance, on AIME24, Qwen3-4B nearly doubles its accuracy from 25.0 to 48.5 with SR-TTRL. Even for the weaker LLaMA3.2-3B-Instruct, our method unlocks a 5-fold improvement (3.8 → 20.0). These results confirm that static inference is far from optimal, and exploring the solution space at test time is a

critical pathway to solving complex reasoning problems.

**Consistent Superiority over Standard TTRL.** When comparing our method against the strong baseline TTRL (Zuo et al., 2025) based on majority voting, SR-TTRL demonstrates consistently superior performance across all five models and four benchmarks. On average, our method provides an additional performance boost ("Δ" in Table 1) ranging from +7.9 to +11.2 over the standard TTRL. Specifically, the gain compared with TTRL on individual benchmark frequently exceeds +10.0, such as the substantial +19.2 advancements observed on the AMC23 for LLaMA3.1-8B-Instruct. These results prove that shifting from quantitative statistical consensus to qualitative logical verification significantly and universally enhances the self-improvement effect, establishing a much higher performance floor for test-time reinforcement learning.

In addition, we conclude the following five phenomena:

**(1) Superior Gains on Challenging Benchmarks.** We observe striking improvements on the challenging benchmarks that require deep and multi-step reasoning. On AIME, our SR-TTRL consistently surpasses standard TTRL by substantial margins. For example, SR-TTRL achieves a +12.7 gain over TTRL for Qwen3-4B model on AIME24, while on LLaMA3.1-8B-Instruct, the margin reaches +14.2. These results demonstrate that when reasoning tasks become sufficiently complex, traditional majority-based TTRL hits a performance bottleneck that can be breached by our method driven by logical verification.

**(2) Thriving in "Logical Minority" Problems.** The performance gap between our SR-TTRL and the standard TTRL is highly correlated with the prevalence of the "logical minority" phenomenon. As illustrated in Appendix Figure 5, on benchmarks like MATH-500 where the gap between Pass@$N$ and Maj@$N$ is relatively small, the correct answer usually occupies the majority, leaving less room for SR-TTRL to differentiate itself from standard TTRL (with gains ranging from +2.6 to +8.1). In contrast, on AIME benchmarks where nearly half of the solvable problems fall into the logical minority category, SR-TTRL thrives by successfully identifying correct solutions that are statistically overshadowed, with gains ranging from +6.5 to +14.2. This alignment confirms that SR-TTRL specifically addresses the limitation of majority voting in scenarios where correct solutions are statistically rare but logically sound. The strength of our method lies in its ability to resolve the distribution shift from unimodal consensus to skewed false-consensus.

**(3) Better Exploitation of Model Capability.** SR-TTRL is more effective at exploiting the inherent capabilities of models. A compelling case is observed in the comparison between LLaMA3.2-3B-Instruct and LLaMA3.1-8B-Instruct. Despite having similar static baseline performance, standard TTRL provides diminishing returns as the model scales from

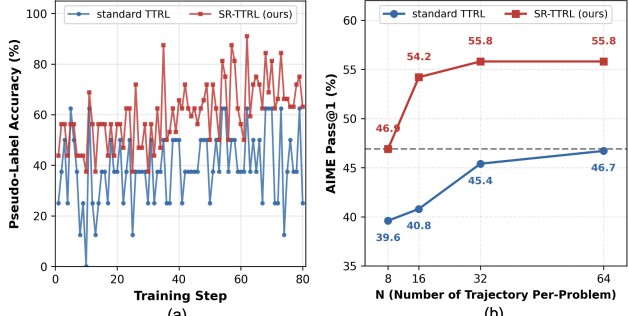

*Figure 4.* **(a)** The evolution of pseudo-label accuracy (vs. ground-truth) during training. **(b)** The effect of sampled trajectory number $N$. All the results are from Qwen3-8B on AIME24.

3B to 8B, with average gains dropping from +9.5 to +7.9 (and from +9.5 to +5.4 on AIME24). This anomaly arises because standard TTRL is strictly bottlenecked by the majority statistical consensus. As shown in Appendix Figure 5, while scaling from 3B to 8B increases the Pass@64 on AIME24 by a substantial +10.0, the Maj@64 only improves by a marginal +3.3. Because TTRL cannot learn from correct trajectories within the logical minority, it fails to capitalize on the 8B model's expanded search space. In contrast, SR-TTRL flourishes by leveraging both the higher generative potential and the enhanced self-reflective depth of the larger 8B model. Consequently, our method yields clearly larger gains for the 8B variant than the 3B variant (+19.1 vs. +17.8 on average; +19.6 vs. +16.2 on AIME24). This confirms that SR-TTRL effectively breaks the statistical ceiling, translating a model's intrinsic reasoning potential into effective test-time learning signals, whereas standard TTRL remains bottlenecked by statistical frequency.

**(4) Superior Sample Efficiency (Quality over Quantity).** Figure 4(b) investigates the impact of the sampled trajectories number $N$ during test-time RL. Surprisingly, SR-TTRL with $N = 8$ achieves comparable performance to TTRL with $N = 64$, indicating the high sample efficiency of our method. For standard TTRL, increasing $N$ from 8 to 64 primarily provides more training samples rather than improving pseudo-label accuracy (since Maj@$N$ stabilizes with $N$). In contrast, SR-TTRL compensates for the smaller sample size with higher pseudo-label precision derived from logical verification. This indicates that improving the quality of pseudo-label is a more sample-efficient strategy for test-time RL than merely increasing the trajectory quantity.

**(5) Higher-Quality Pseudo-Labels.** As shown in Figure 4(a), SR-TTRL consistently generates pseudo-labels with substantially higher accuracy than standard TTRL throughout training. The red curve (SR-TTRL) maintains an average pseudo-label accuracy of ∼0.65, with frequent peaks above 0.7 and even reaching near-perfect accuracy (more than 0.9) at several steps. In contrast, the blue curve (standard

*Table 2.* Ablation on the effect of each step in Self-Reflective Pseudo-Labeling Mechanism (in Sec. 4.2).

| Candidate Trajectory Pool Construction | Trajectory Summarization | Self-Reflective Verification | AIME24 |
|:---:|:---:|:---:|:---:|
| ✓ | ✓ | ✓ | **55.8** |
| - | ✓ | ✓ | 54.3 |
| ✓ | - | ✓ | 50.4 |
| - | - | ✓ | 48.8 |

*Table 3.* Comparison of different pseudo-labeling signals on AIME24 using Qwen3-8B. "Ground-truth" represents the upper bound of test-time reinforcement learning.

| Pseudo-Label Signal | Pass@1 |
|:---|:---:|
| Ground-truth | **65.4** |
| Self-Reflective (in our SR-TTRL) | 55.8 |
| Majority-Voted (in standard TTRL Baseline) | 46.7 |

*Table 4.* Speed comparison between the standard TTRL and our SR-TTRL. The results are tested on Qwen3-8B trained on the AIME24 benchmark.

| Method | Time per step | Pass@1 |
|:---:|:---:|:---:|
| TTRL | **160** seconds | 46.7 |
| SR-TTRL | 205 seconds | **55.8** |

TTRL) exhibits lower quality and larger fluctuations, frequently dropping below 0.3. Crucially, SR-TTRL achieves its advantage from the very first step. This demonstrates that self-reflection enables robust pseudo-labeling, while majority voting remains fragile and unstable due to its dependence on statistical dominance. The sustained quality gap directly explains SR-TTRL's superior final performance: high-fidelity rewards from early stages accelerate learning and prevent policy collapse during adaptation.

### 5.3. Ablation Study

**Effect of Key Components.** Table 2 ablates the two core steps in our self-reflective pseudo-labeling mechanism. When "Candidate Trajectory Pool Construction" is not used, we explicitly add the instruction *"Focus solely on the logical rigor of the skeletons, disregarding the frequency of any specific answer"* to the self-reflective verification prompt. This ensures a fair comparison by preventing the model from being biased toward frequent answers due to input redundancy. Despite this safeguard, performance still drops from 55.8 to 54.3, indicating that redundant trajectories, even when frequency is ignored, dilute the reflection process and impair selection quality. Disabling "Trajectory Summarization" (*i.e.,* using full-length trajectories) causes a larger drop to 50.4, due to context overflow and increased noise. The full degradation to 48.8 when both components are removed confirms that candidate selection and concise reasoning representation are essential for effective self-reflection.

**Impact of Sample Size $N$.** As Figure 4(b), the performance of SR-TTRL increases rapidly with the number of sampled trajectories $N$ and then saturates after $N = 32$, thus we set $N = 32$ by default.

### 5.4. Further Analysis

**Proximity to the Performance Upper Bound.** To further evaluate the effectiveness of our self-reflective pseudo-labeling, we compare SR-TTRL against an Oracle training scenario (using ground-truth labels for all sampled trajectories) and the standard Majority Voting baseline. The Oracle performance represents the upper bound for test-time reinforcement learning, as it assumes a perfect reward signal that identifies every correct solution within the sample pool. As shown in Table 3, SR-TTRL significantly narrows the gap between statistical consensus and the theoretical perfor-

mance ceiling. While majority voting achieves 46.7, leaving an 18.7 gap to the Oracle, SR-TTRL reaches 55.8, effectively reducing this gap to only 9.6. This means that our method recovers nearly half of the "lost potential" that statistical methods fail to capture. The results demonstrate that the primary bottleneck in conventional TTRL is not the model's inability to learn, but rather the noise and unreliability of the reward signal. By providing a higher-fidelity signal, SR-TTRL allows the model to approach its true reasoning limit more closely than any frequency-based method.

**Speed Analysis.** We measure the average time per training step and final Pass@1 accuracy on AIME24 using Qwen3-8B. As shown in Table 4, standard TTRL takes 160 seconds per step and achieves 46.7 Pass@1, while SR-TTRL requires 205 seconds per step (a 28 increase in latency) but attains a substantially higher 55.8 Pass@1, with +9.1 gain. This demonstrates that the additional computational cost of self-reflective pseudo-labeling is well justified by the significant improvement in reasoning ability on challenging problems.

## 6. Conclusion

In this paper, we introduced Self-Reflective Test-Time Reinforcement Learning (SR-TTRL), a novel framework that addresses the fundamental limitation of majority voting in complex reasoning tasks. By shifting from statistical consensus to logical self-reflection, SR-TTRL groups, summarizes, and critically evaluates diverse reasoning trajectories to generate high-fidelity pseudo-labels without ground-truth supervision. Extensive experiments across various model families and benchmarks validate the effectiveness of our approach. Notably, SR-TTRL boosts the Pass@1 accuracy of Qwen3-8B on the challenging AIME24 benchmark by +26.7%, outperforming standard TTRL by a wide margin. Our findings demonstrate that internal verification is a powerful alternative to majority-driven heuristics, paving a new way for self-evolving autonomous agents at test time.

## Acknowledgements

This work was supported in part by the Research Grants Council under the Areas of Excellence scheme grant AoE/E-601/22-R. The work has been supported by Hong Kong Research Grant Council-General Research Fund Scheme (Grant No. 17202422, 17212923, 17215025), Theme-based Research (Grant No. T45-701/22-R), and Strategic Topics Grant (Grant No. STG3/E-605/25-N). Part of the described research work is conducted in the JC STEM Lab of Robotics for Soft Materials, funded by The Hong Kong Jockey Club Charities Trust.

## Impact Statement

This work aims to improve the self-improvement capability of large language models on complex reasoning tasks through test-time adaptation. By enhancing a model's ability to reflect on and learn from its own reasoning processes, our method could contribute to more reliable and autonomous AI systems for domains such as education, scientific exploration, and decision support. However, like any advancement in AI reasoning, it should be deployed with appropriate safeguards to ensure outputs are verified by human experts in high-stakes applications.

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

## Limitations

While our approach has demonstrated significant progress in math reasoning tasks, this study is subject to certain limitations due to constraints in computational resources and time. Specifically, we have yet to conduct extensive experimental validation in broader scenarios such as Agentic learning and Multimodal learning. We plan to extend our evaluation to these domains in future work to further verify the generalizability and effectiveness of our methods.

## LLM-Usage Statements

The authors confirm that AI tools were utilized solely for identifying typographical errors and performing minor linguistic polishing to improve readability. No part of the core research, data analysis, or substantive writing was generated by a Large Language Model (LLM).

## A. Prompt Templates

We provide the exact prompt templates used in our Self-Reflective Pseudo-Labeling mechanism (Sec. 4.2). The prompts are carefully designed to guide the model toward precise, logic-focused behaviors while avoiding common pitfalls such as verbosity, redundancy, or frequency bias. Both prompts avoid open-ended instructions and instead provide concrete, actionable criteria. This specificity is crucial for eliciting consistent, high-quality self-reflection from the model.

### A.1. Prompt for Trajectory Summarization

```
Prompt for Trajectory Summarization

Given a problem and its corresponding solution, your task is to condense the long reasoning trajectory into
a concise reasoning skeleton while preserving its deductive integrity.

To achieve this, please distill the content according to the following requirements:
1. Retain all key mathematical derivations, equations, and intermediate numerical results that are essential
to the logical flow.
2. Retain critical decision points and logical transitions.
3. Remove all auxiliary explanations, conversational fillers, and stylistic verbiage.
4. Remove invalid exploration paths, erroneous steps that were later self-corrected, and unimportant
intermediate details (e.g., overly granular algebraic expansions or repetitive computations).
5. Express the retained content using the most concise yet precise language possible, avoiding redundancy
without sacrificing clarity.
6. Do not correct or modify any part of the original reasoning, even if you detect errors. Your role is to
summarize faithfully, not to fix.

Problem:
{problem}

Original Trajectory:
{trajectory}

Summarized Reasoning Skeleton:
```

The "Trajectory Summarization" prompt instructs the model to distill a full-length reasoning trajectory into a minimal yet logically complete "reasoning skeleton." Key design choices include:

- ***Preserve Core Logic:*** Explicitly requiring retention of "key mathematical derivations and intermediate results" to preserve deductive integrity;

- ***Prune Irrelevant Exploration:*** Mandating removal of "self-corrected errors, invalid or unimportant exploration paths", which are common in raw LLM outputs but irrelevant to final correctness;

- ***Standardize Output Format:*** Enforcing a standardized output format with the final answer enclosed in `\\boxed{}` to ensure compatibility with downstream processing;

- ***No Explicit Length Constrain:*** Avoiding a hard token limit, as the compressible length varies significantly with problem complexity, imposing a fixed bound could truncate essential steps in harder problems or force unnecessary padding in simpler ones.

This flexible compression strategy enables efficient comparison within context window constraints while preserving critical reasoning content across diverse problem difficulties.

In addition, we provide the following explanation for the items in "Requirement 3" for clarity:

- *Auxiliary explanations*: Redundant justifications or restatements that reiterate already-established facts (e.g., "As we saw earlier..." or "This is because...") without advancing the core logic;

- *Conversational fillers*: Phrases that mimic human dialogue but add no mathematical value (e.g., "Let's think step by step," "Hmm, I wonder...," or "Okay, so...");

- *Stylistic verbiage*: Decorative language, rhetorical questions, or overly verbose phrasing intended to sound fluent rather than precise (e.g., "It is worth noting that..." or "A careful observer might realize...").

By targeting these specific sources of noise, the summarization process focuses exclusively on the logical backbone of the solution, ensuring that the resulting skeleton is both compact and faithful to the original reasoning.

### A.2. Prompt for Self-Reflective Verification

---

**Prompt for Self-Reflective Verification**

```
Given a problem and {G} distinct candidate reasoning skeletons, your task is to verify the logical soundness
of each candidate and identify the one most likely to be correct.

Some candidates may contain subtle calculation errors or logical fallacies. Please perform the verification
based on the following requirements:
1. Compare the different reasoning skeletons side-by-side to identify the most robust logical path.
2. Verify each derivation step for consistency, mathematical accuracy, and soundness.
3. Confirm that the final answer enclosed in \\boxed{} is correctly and necessarily derived from its
preceding steps.
4. Focus solely on the logical rigor of the skeletons.
5. Select the index of the best candidate.

Output Format:
Please first analyze the candidates and output your rationale, and then put your final choice in the format:
``The most reliable candidate is [Candidate k]''.

Problem:
{problem}

Candidates:
[Candidate 1]:
{summared_trajectory_1}

[Candidate 2]:
{summared_trajectory_2}
...
[Candidate {G}]:
{summared_trajectory_{G}}
```

---

The "Self-Reflective Verification" prompt directs the model to act as an impartial evaluator, comparing multiple candidate solutions. Its core features are:

- *Joint comparative evaluation*: Candidates must be analyzed together, not in isolation, to enable direct cross-examination of reasoning paths and identification of subtle logical discrepancies.

- *Step-by-step logical verification*: The model is instructed to scrutinize each derivation for mathematical accuracy, internal consistency, and soundness, rather than making a holistic or intuition-based judgment.

- *Structured output with explicit rationale*: The response format requires a clear justification followed by a definitive selection (e.g., "The most reliable candidate is [Candidate k]"), which discourages arbitrary choices and promotes deliberate, traceable reasoning.

By avoiding vague directives and instead providing concrete, actionable criteria, this prompt ensures that self-reflection is systematic, reproducible, and focused on logical rigor, key prerequisites for high-fidelity pseudo-labeling.

# B. Comprehensive Performance Comparison of Pass@64 vs. Maj@64

To provide a detailed landscape of model capabilities and the limitations of statistical consensus, we report the Pass@64 (generative potential) and Maj@64 (consensus accuracy) across four benchmarks for all evaluated models. As shown in Figure 5, we observe three main phenomena:

**(1) Task-Specific Gap Variance.** The experimental results reveal that the gap between Pass@$N$ and Maj@$N$ is highly sensitive to task difficulty.

- On Hard Benchmarks (AIME24, AIME25), the gaps are consistently massive across all models. For instance, Qwen3-8B achieves an 80.0 Pass@64 on AIME24 but only 43.3 Maj@64, a gap of 36.7.
- On Simpler Benchmarks (MATH-500), the gaps for Qwen models narrow significantly as performance approaches saturation (e.g., Qwen3-8B: 98.4 vs 92.4). However, for LLaMA3.2-3B-Instruct, even on MATH-500, a significant gap of 29.8 (88.2 vs 58.4) persists, indicating that statistical consensus remains a bottleneck for these models even on easier tasks.

**(2) Scaling Divergence in Potential vs. Consensus.** A crucial observation from the LLaMA series results is the divergence between potential and realized performance during scaling. On AIME24, as we scale from LLaMA3.2-3B-Instruct to LLaMA3.1-8B-Instruct:

- The Pass@64 increases by $+10.0$ ($30.0 \rightarrow 40.0$), showing a clear expansion of the model's search space and generative capacity.
- The Maj@64, however, only improves by a marginal $+3.3$ ($10.0 \rightarrow 13.3$).

This quantitative evidence supports our argument in that larger models generate a higher volume of correct solutions that are likely trapped in the "logical minority". While standard TTRL is limited by the stagnant Maj@$N$ signal, our SR-TTRL successfully capitalizes on the 10.0 potential growth, explaining its superior scalability.

**(3) Consistency Across Model Families.** The "Logical Minority" phenomenon is not unique to a specific architecture. Both the Qwen and LLaMA families exhibit the same pattern: high-quality solutions are present in the sampling pool but are frequently outvoted by systematic errors. This universality underscores the fundamental need for qualitative verification mechanisms like SR-TTRL in the test-time training paradigm.

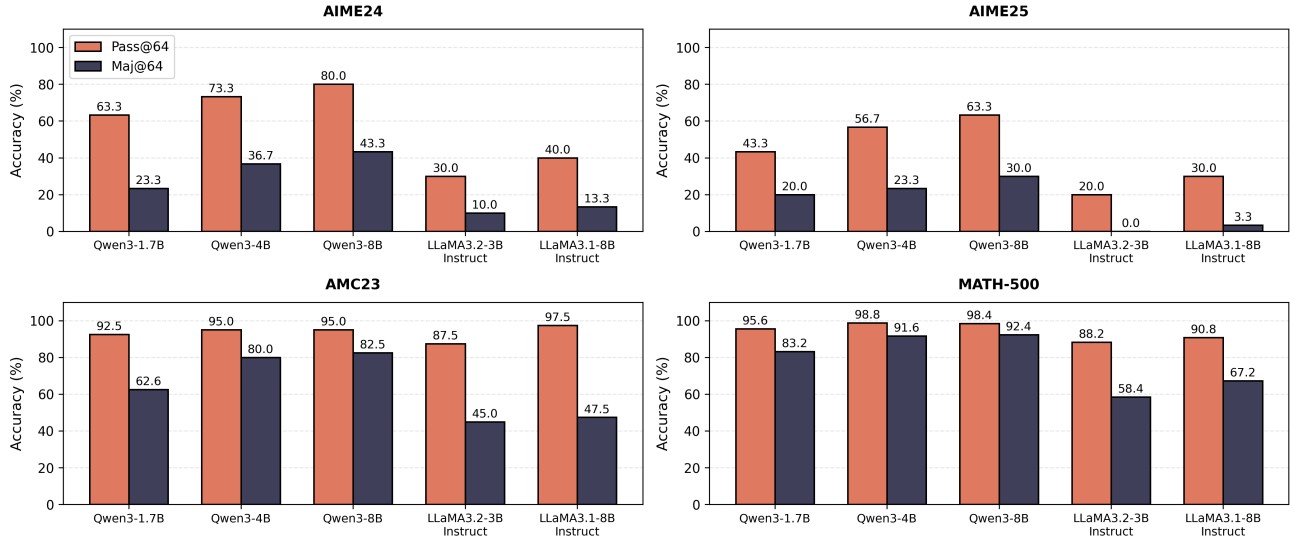

*Figure 5.* The Pass@64 and Maj@64 performance across four benchmarks for five models.

