# OpenReview forum: "Beyond Majority Voting: Self-Reflective Test-Time Reinforcement Learning for LLM Reasoning"
_ICML.cc/2026/Conference — ICML 2026 regular_

### Official Review · Reviewer_DiTj · 2026-03-06

**Soundness:** 3
**Presentation:** 2
**Significance:** 2
**Originality:** 2
**Overall Recommendation:** 4
**Confidence:** 3

**Summary:**

This paper proposes to use logical minority instead of majority vote as the target for RL training when the gold label is not present. The authors observe a performance gap between pass@k and maj@k for reasoning models, suggesting majority voting doesn't always identify the best answer. Their method generates higher-quality pseudo-labels by: (1) grouping rollouts by their final answers, (2) selecting a representative from each group, (3) summarizing them, and (4) having the model judge which is most logically sound. Experiments show this outperforms TTRL and narrows the gap with gold-label training.

**Compliance With Llm Reviewing Policy:**

Affirmed.

**Ethical Review Concerns:**

No ethical concerns.

**Final Justification:**

I have increased my score after reading the author response.

**Key Questions For Authors:**

- The paper uses the same benchmarks (AIME24, AIME25, AMC23, MATH-500) for both training and evaluation. Have you evaluated whether SR-TTRL generalizes to held-out test sets when trained on a separate dataset (e.g., NuminaMath/DAPO)? If not, how do you ensure the improvements reflect genuine reasoning gains rather than in-distribution adaptation?
- Why were Qwen3 models evaluated in non-thinking mode (Line 690)? Given that Qwen3-4B achieves 73.8 on AIME24 in thinking mode versus 25.0 in non-thinking mode, the practical relevance of improving non-thinking models is unclear. If thinking-mode instruction-tuned models are difficult to improve due to proprietary alignment data, have you considered starting from a base model (e.g., Qwen3-8B-Base) and enabling thinking mode during SR-TTRL training?

**Limitations:**

yes

**Strengths And Weaknesses:**

## Presentation

The paper is generally well-written and easy to follow. However, the balance between motivation and experimentation feels skewed. The phenomenon of "logical minority" is discussed extensively, while the core experimental evaluation spans only ~2 pages. I would encourage the authors to expand the experimental analysis, as this is where the contribution is ultimately validated.

Additionally, moving critical details such as training and evaluation protocols to the Appendix may cause distractions to the readers.

## Soundness

I have several concerns about the experimental setup:

**(1) Train-Test Contamination.** The paper uses the same benchmarks (AIME24, AIME25, AMC23, MATH-500) for both training and evaluation. While this is inherent to the TTRL paradigm, it raises questions about generalization. A stronger evaluation would train on a separate, larger-scale dataset (e.g., NuminaMath/DAPO dataset) and evaluate on held-out test sets like AIME24/25. This would better demonstrate whether SR-TTRL improves generalization rather than just in-distribution adaptation.

**(2) Non-Thinking Mode Evaluation.** Line 690 states that "Qwen3 models are evaluated in non-thinking mode," which is a significant concern. According to Qwen3's official report, thinking mode achieves 73.8 on AIME24 for Qwen3-4B, far exceeding the non-thinking baseline (25.0) and even the SR-TTRL result (48.5). This raises two issues:
   - Why evaluate in non-thinking mode when the target application is improving reasoning models?
   - Does SR-TTRL scale to thinking mode, where trajectories are substantially longer and more complex? It is unclear whether the summarization and self-reflection mechanisms can handle such outputs effectively.

Without experiments on thinking-mode models, the practical applicability of this work to state-of-the-art reasoning systems remains uncertain. If the authors' concern is that instruction-tuned thinking models are difficult to improve upon due to proprietary alignment data, a reasonable alternative would be to start from a base model (e.g., Qwen3-4B-Base) and enable thinking mode during training and evaluation. This would provide a fairer testbed while still evaluating the method on the reasoning regime it is designed for.

## Significance

Due to the concerns raised above—particularly regarding evaluation protocols and the choice of non-thinking mode—the significance of this work is limited. If addressed, the contribution would be more compelling.

## Originality

The idea of replacing majority voting with self-reflective verification is intuitive and well-motivated. However, the landscape of TTRL improvements is evolving rapidly. The paper would benefit from discussing and comparing against works such as Self-Harmony[1], RESTRAIN [2], SCOPE [3] etc. that are closely relevant.

---
[1] Self-Harmony: Learning to Harmonize Self-Supervision and Self-Play in Test-Time Reinforcement Learning

[2] RESTRAIN: From Spurious Votes to Signals -- Self-Driven RL with Self-Penalization

[3] Beyond Majority Voting: Towards Fine-grained and More Reliable Reward Signal for Test-Time Reinforcement Learning

---

> ### Author Rebuttal · Authors · 2026-03-31
>
> ## **Q1: Generalizability Analysis**
> We train Qwen3-1.7B on DAPO-17k and evaluate on AIME. The results (Table below) show that SR-TTRL maintains a significant advantage, proving that the self-reflective mechanism is not limited to in-distribution adaptation. We will incorporate these results into the revision to further demonstrate the generalization capabilities of our method.
>
> ||AIME24|AIME25|
> |:---|:---|:---|
> |Qwen3-1.7B-NonThinking|13.4|9.8|
> |+TTRL|18.8|17.9|
> |+SR-TTRL|26.7|22.1|
> |Qwen3-1.7B-Thinking|48.3|36.8|
> |+TTRL|50.4|39.2|
> |+SR-TTRL|54.2|42.9|
>
> ---
> ## **Q2: Results for Qwen3-models under Thinking Mode**
> We supplement the experiments for Qwen3-1.7B and 8B, training and evaluation under "thinking mode".
>
> The format of trajectory under thinking mode is: `<think> thinking process </think> <answer> solution </answer>`. In our Step 1 (Candidate Construction), we use the solution part extracted from the `<answer>` tags as the candidate trajectory. In Step 2 (Summarization), we use the model in non-thinking mode to generate summary, because using thinking mode in this step does not yield gains. In Step 3 (Verification), we enable thinking mode for the self-reflective verification to allow the model to deliberate deeply on the logical integrity of each candidate.
>
> The results are summarized below:
>
> ||AIME24|AIME25|
> |:---|:---|:---|
> |Qwen3-1.7B-Thinking|48.3|36.8|
> |TTRL|51.3|40.8|
> |SR-TTRL|56.2|43.3|
> |Qwen3-8B-Thinking|76.0|67.3|
> |TTRL|76.3|70.4|
> |SR-TTRL|78.3|75.0|
>
> **Why SR-TTRL Scales to Thinking Models:**
> - **Higher Ceiling:** Thinking-mode models exhibit a significantly higher Pass@N potential. This provides a much "richer" candidate pool for SR-TTRL, effectively raising the upper bound of pseudo-label accuracy.
> - **Stronger Discriminator:** Enabling thinking mode in Step 3 (Verification) allows the model to perform more granular logical analysis. This directly translates to higher verification competence, enabling the model to more reliably distinguish the "logical minority" from systematic errors.
>
> These results confirm that SR-TTRL is highly effective for state-of-the-art reasoning models and effectively utilizes their expanded search space and deeper reflection capabilities. We will include this analysis in the revised manuscript.
>
> ---
> ## **Q3: Comparison with More Baselines**
> **(1) Comparison with Self-Harmony:**
>
> Due to the character constraints of the rebuttal, we kindly invite the reviewer to refer to the detailed analysis in our response to Reviewer h9BT Q1.
>
> **(2) Comparison with RESTRAIN:**
>
> **Method:**
> - RESTRAIN mitigates the fragility of majority voting through pseudo-label soft weighting, negative rollout penalization, and prompt-level weighting. However, its core mechanism remains frequency-driven, still biased toward high-frequency answers, unable to fundamentally identify correct solutions within the "logical minority."
> - In contrast, SR-TTRL achieves a paradigm shift: from statistical frequency to logical quality verification, which enables the model to directly compare the reasoning rigor of multiple candidate trajectories, thereby identifying "logical minority" solutions that are infrequent but logically sound.
> - The essential difference between these two approaches lies in their core philosophy: RESTRAIN attempts to "smooth" the frequency distribution to avoid the fragility of majority voting, whereas our SR-TTRL "transcends" frequency itself by directly evaluating reasoning quality.
>
> **Experiments:** We conduct fair comparisons on Llama3.1-8B-Instruct (RESTRAIN results taken from original paper Table 3).
> ||AIME24|AMC23|MATH-500|
> |:---|:---|:---|:---|
> |+TTRL|10.0|32.3|63.7|
> |+RESTRAIN|16.7|40.0|67.4|
> |+SR-TTRL|24.2 (+7.5)|51.5 (+11.5)|72.8 (+5.4)|
>
> **(3) Comparison with SCOPE:**
>
> **Method:**  SCOPE enhances majority voting through step-wise confidence weighting aggregation, but remains frequency-driven. It improves how frequencies are aggregated, not whether frequency should be the selection criterion. In contrast, SR-TTRL represents a paradigm shift from statistical consensus to logical consensus. Our self-reflective mechanism enables the model to simultaneously observe multiple candidate trajectories and perform joint comparative verification. This "contrast-driven" evaluation mechanism is more effective at unlocking the model's intrinsic reasoning potential than "confidence-driven," single-trajectory assessments, which brings two critical advantages:
> - Overcoming "Confident Errors"
> - Stimulating Deep Reflection: Presenting multiple reasoning paths side-by-side creates strong cognitive contrast, prompts the model to proactively identify logical discrepancies and uncover reasoning flaws.
>
> **Experiments:** We conduct fair comparisons on Qwen3-1.7B (SCOPE results taken from original paper Table 1), showing clear advantages of our method.
> ||AIME24|AMC23|MATH-500|
> |:---|:---|:---|:---|
> |+TTRL|19.8|50.2|84.9|
> |+SCOPE|21.7|53.5|81.3|
> |+SR-TTRL|26.3 (+4.6)|66.5 (+13.0)|89.3 (+8.0)|

---

> > ### Author Rebuttal · Reviewer_DiTj · 2026-04-04
> >
> > Thanks for the author response. I will increase my score.

---

> > > ### Author Response · Authors · 2026-04-04
> > >
> > > Dear Reviewer DiTj,
> > >
> > > We are pleased to hear that our responses have addressed your concerns. We sincerely thank you for the positive feedback and score increase.
> > >
> > > Your insightful suggestions have significantly enhanced the quality and clarity of our manuscript. We are committed to incorporating all additional results and discussions into the final version of our paper.
> > >
> > > Thank you again for your valuable time and constructive feedback.
> > >
> > > Best Regards,
> > >
> > > Paper 11022 Authors

---

### Official Review · Reviewer_mZVG · 2026-03-11

**Soundness:** 2
**Presentation:** 3
**Significance:** 2
**Originality:** 3
**Overall Recommendation:** 4
**Confidence:** 4

**Summary:**

This paper takes a close look at a limitation in test-time reinforcement learning, particularly in how pseudo-labels are constructed. In standard TTRL setups, majority voting is often used to determine the target answer among multiple sampled reasoning trajectories. However, as the authors point out, this approach can systematically suppress reasoning paths that are in the minority yet actually correct. I find this observation both sharp and practically relevant, especially in settings where correct reasoning may not be the most frequent outcome.

To address this issue, the paper introduces the notion of a “Logical Minority” and proposes the SR-TTRL algorithm. Instead of relying on frequency, the method evaluates the logical quality of representative trajectories corresponding to different answers and selects the most reliable one. In other words, the model is encouraged to compare and reflect on alternative reasoning paths, rather than simply following the majority. Conceptually, I find this shift away from naive voting toward self-evaluated reasoning quite appealing.

Empirically, the authors evaluate SR-TTRL on four challenging benchmarks: AIME24, AIME25, AMC23, and MATH-500. Experiments are conducted across multiple base models, and SR-TTRL is compared directly against standard majority-voting-based TTRL. The results consistently show that SR-TTRL outperforms the baseline across models and benchmarks. These findings suggest that enabling models to reflect on and compare different reasoning trajectories can be more effective than majority voting in extracting correct answers and improving test-time reinforcement learning performance.

**Compliance With Llm Reviewing Policy:**

Affirmed.

**Final Justification:**

My concerns have been addressed, and I have raised my score.

**Key Questions For Authors:**

1. I would be interested to see how SR-TTRL performs on more general benchmarks beyond math-oriented reasoning tasks. For example, how does it behave on MMLU-Pro? More broadly, does SR-TTRL exhibit meaningful generalization ability outside the relatively narrow distribution of mathematical reasoning problems considered in the current paper?

2. Could the authors provide a more explicit accounting of the overhead introduced by SR-TTRL, including grouping, summarization, and self-reflective comparison? A quantitative breakdown of these additional costs would make it much easier to judge the true efficiency of the method in practice.

**Limitations:**

Please refer to the weaknesses discussed above, which outline the main limitations of the proposed method, including the ambiguity of phase definition, indirect and potentially unstable routing supervision, possible sample inefficiency due to isolated expert updates, limited routing capacity, lack of knowledge sharing across experts, additional computational overhead, and the absence of rigorous theoretical analysis.

**Strengths And Weaknesses:**

## Strengths

1. I find the motivation of this paper particularly clear and, frankly, quite insightful. Much of the current literature focuses on improving reasoning ability itself, often by scaling models or refining training strategies. However, this work points out that a core bottleneck in TTRL may actually lie in the quality of pseudo-labels. In my view, identifying majority voting as a hidden failure mode in complex reasoning tasks is a meaningful contribution. The articulation of the “Logical Minority” phenomenon feels both intuitive and overdue.

2. The proposed SR-TTRL method is conceptually simple and easy to implement. I appreciate that the authors do not introduce additional large models or complicated architectural changes. Instead, they replace the pseudo-label generation component within the existing TTRL framework. Despite this relatively lightweight modification, the performance gains are quite strong. To me, this simplicity is part of the appeal, as it suggests the idea is robust rather than over-engineered.

3. The experimental results provide convincing support for the method. SR-TTRL is evaluated across different base models and multiple benchmarks, and it consistently outperforms standard TTRL. The improvements are particularly noticeable on more challenging tasks, which strengthens the claim that majority voting becomes unreliable in harder reasoning scenarios. I find the central message compelling: encouraging the model to reflect on and compare alternative reasoning trajectories appears to extract correct answers more effectively than relying on frequency alone.

4. The paper is also well structured and clearly written. The narrative flows naturally from the observed phenomenon to the proposed method. As a reader, I found it easy to follow the reasoning behind each design choice. The logical progression from problem identification to algorithmic solution is, in my opinion, one of the strengths of the presentation.

## Weaknesses

1. I am concerned that the effectiveness of SR-TTRL depends quite heavily on the model’s existing verifier or self-evaluation capability. The central assumption seems to be that the model is better at comparing and reflecting over candidate reasoning paths than at directly generating a high-quality reasoning trajectory in the first place. I am not fully persuaded that this assumption holds in general, especially for weaker base models. The paper does not offer convincing empirical or theoretical validation of this premise. As a result, I remain unconvinced that SR-TTRL would consistently deliver improvements when the underlying model has limited reflective strength.

2. In the current setup, the same model is responsible for generating reasoning trajectories and performing self-reflection over them. I find this somewhat problematic. If the model’s reasoning generation is weak, biased, or systematically flawed, those issues are very likely to reappear in the reflection stage. Moreover, the reflection process involves non-trivial sub-tasks such as summarization and structured comparison, which themselves may require substantial reasoning capability. Prior work, such as multi-agent debate [1] and uncertainty-aware reasoning frameworks [2], suggests that stronger or external verifiers can significantly impact reasoning quality. I would have strongly preferred to see an analysis of how SR-TTRL performs when paired with a stronger verifier, or at least a clearer discussion of how sensitive it is to the underlying model’s reflective ability.

3. SR-TTRL relies on summarization rather than comparing full reasoning trajectories. While I understand the computational motivation, I am somewhat skeptical about the reliability of this design choice. The summarization step inevitably compresses tokens, and in doing so may remove crucial intermediate reasoning details. Important logical steps that determine correctness could be omitted or distorted. The paper does not thoroughly investigate whether this information loss meaningfully affects the subsequent comparison stage, which leaves me uncertain about the robustness of the approach.

4. Although the paper emphasizes improved sample efficiency, I find it somewhat incomplete that the additional overhead introduced by SR-TTRL is not carefully quantified. The pipeline includes grouping, summarization, and reflective comparison, all of which incur extra computational cost. It is not entirely clear to me whether the overall approach remains cost-effective in realistic deployment scenarios. I would have liked to see a more transparent accounting of the total compute and token overhead before accepting the efficiency claims at face value.

5. The performance gains of SR-TTRL are supported primarily by empirical results. While the intuition behind logical-minority-based selection is appealing, I am not fully convinced that the paper provides a sufficiently deep theoretical explanation for why it should systematically outperform majority voting under broad conditions. A stronger theoretical argument would make the claims more persuasive.

6. All experiments are conducted on mathematics-oriented reasoning benchmarks. While this is a reasonable testbed, I am not yet convinced that the method generalizes beyond this domain. It would be important to evaluate SR-TTRL on more general multi-task reasoning benchmarks such as MMLU-Pro [3]. Without such evidence, I find it difficult to assess the breadth of applicability of the proposed approach.

7. Finally, the paper does not report standard deviations or other measures of statistical uncertainty. Given the variability often observed in reinforcement learning and reasoning experiments, I would have strongly preferred to see results averaged over multiple random seeds. Without this information, it is harder to judge the robustness of the reported improvements.
---

### References
- [1] Du, Yilun, et al. "Improving factuality and reasoning in language models through multiagent debate." Forty-first international conference on machine learning. 2024.
- [2] Wang, Ruhan, et al. "FERA: Uncertainty-aware Federated Reasoning for Large Language Models."
- [3] Wang, Yubo, et al. "Mmlu-pro: A more robust and challenging multi-task language understanding benchmark." Advances in Neural Information Processing Systems 37 (2024): 95266-95290.

---

> ### Author Rebuttal · Authors · 2026-03-31
>
> ## **Q1: Clarification on Self-Reflective Capability**
> - **Theoretical Intuition: Selection vs. Generation**
> The task of generation requires the model to navigate an astronomical search space of token sequences to find a single valid path. In contrast, SR-TTRL's verification task is reduced to a multiple-choice selection over a small, deduplicated candidate pool $\mathcal{P}$ (where $|\mathcal{P}| = G$, typically $G \ll N$).Mathematically, as long as the model's verification competence ($\gamma$) is higher than the raw generation probability ($p(a^*)$), SR-TTRL will outperform majority voting. Given that the verifier only needs to identify logical inconsistencies in a summarized "skeleton," the cognitive load is significantly lower than generating a full-length proof from scratch.
>
> - **Empirical Validation on Weaker Models**: Even small-scale models achieve substantial gains in Table 1. For instance, LLaMA-3.2-3B improved from 3.8 to 20.0 (+16.2) on AIME24, and Qwen3-1.7B saw an average +14.2% improvement.
> ---
> ## **Q2: Experiments on External Verifiers**
> We conduct experiments using stronger external models as the verifier in step 3 of SR-TTRL pipeline:
> |Pseudo-Label|Qwen3-8B on AIME24|
> |:---|:---|
> |Ground-Truth|65.4|
> |Maj-Vote|46.7|
> |Self-Reflective (self-verifier)|55.8|
> |Self-Reflective (Qwen3-32B verifier)|56.7|
> |Self-Reflective (GPT-5.2 verifier)|60.4|
>
> - **Scaling with Verifier Strength**: Accuracy improves as the verifier becomes stronger, narrowing the gap to the Oracle.
> - **Effectiveness of Self-Verification**: Notably, the 8B self-verifier already captures over **50%** of the potential gain between Maj-Vote and the Oracle. This confirms that even the base model possesses sufficient reflective capacity to identify the "logical minority".
>
> While external verifiers offer higher ceilings, our results demonstrate that self-verification is a viable and powerful strategy for autonomous self-evolution. We will include this sensitivity analysis in the revision.
>
> ---
> ## **Q3: Reliability and Necessity of Summarization**
> **(1) Reliability Analysis:**
> We use LLM-as-a-judge (GPT-5.2) as an impartial judge to evaluate whether the summarized "reasoning skeletons" accurately preserve the logical error steps of the raw trajectories. Around 95.7% of logical error steps from the raw trajectories were accurately retained in the summarized skeletons. Thus, summarization is highly faithful.
>
> **(2) Empirical Impact: Noise vs. Minor Information Loss.**
> Our experiments confirm that while summarization involves a minor compression of tokens, the noise reduction it provides is indispensable for effective verification. Disabling summarization causes a performance drop due to increased stylistic noise and context overflow, which distracts the model from core logical rigor. We compare the pseudo-label fidelity and test benchmark accuracy in Table below:
> | Method | Pseudo-Label Accuracy | AIME24 |
> | :--- | :--- |:--- |
> |Pass@64|80.0|65.4|
> |Maj-Vote|43.3|46.7|
> |Self-Reflective|66.7|55.8|
> |Self-Reflective w/o Summarization|60.0|50.4
>
> **Conclusion:** The minor information loss is statistically negligible compared to the significant gains achieved by providing the verifier with a clean, logic-focused reasoning skeleton. The primary bottleneck remains the model's Verification Competence, not the summarization step.
>
> ---
> ## **Q4: Computational Cost**
>
> **(1) Overhead Breakdown:** As Table 4 in the paper, the total time per training step is 160s for standard TTRL and 205s for SR-TTRL. The additional 45s overhead is distributed as follows:
>
> ||Time per step||
> |:---|:---:|:---|
> |TTRL|160s|
> |SR-TTRL|205s||
> |--- *Breakdown of +45s* ---||||
> |Step 1 (Candidate Trajectory Pool Construction)|<1s|CPU-based, negligible cost|
> |Step 2 (Trajectory Summarization)|26s| Parallel inferences. Since the generated "skeletons" are significantly shorter than full trajectories, the token overhead is minimal |
> |Step 3 (Self-Reflective Verification)|19s||
>
> **(2) Token Efficiency and Deployment Cost-Effectiveness:** While SR-TTRL adds reflection overhead per step, it significantly improves sample efficiency, which is the primary driver of total compute in TTRL. SR-TTRL (N=8) reaches identical performance with TTRL (N=64), resulting in a 2.1x speed acceleration (132s vs 278s).
>
> ---
> ## **Q5: Theoretical Analysis**
> Due to character constraints, we kindly invite the reviewer to refer to the detailed theoretical analysis in our response to Reviewer h9BT Q2.
>
> ---
> ## **Q6: Additional Experiments on Other Benchmarks (science, puzzle, coding, general reasoning)**
> Due to character constraints, we kindly invite the reviewer to see the results (with obvious gains) in our response to Reviewer f9WW Q6.
>
> ---
> ## **Q7: Statistical Uncertainty**
> We conduct 3 independent runs for Qwen3-1.7B on AIME24 using different random seeds. The results are 26.3, 26.3, and 26.7 (good robustness). We will include averaged results and standard deviations for all experiments in the revision.

---

> > ### Author Rebuttal · Reviewer_mZVG · 2026-04-04
> >
> > Thank you for your response. My concerns have been addressed, and I have raised my score.

---

> > > ### Author Response · Authors · 2026-04-04
> > >
> > > Dear Reviewer mZVG,
> > >
> > > We are pleased that your concerns have been fully addressed. We sincerely thank you for your positive evaluation and for raising the score.
> > >
> > > Your valuable suggestions have significantly strengthened the quality and clarity of our work. We commit to incorporating the additional results and analysis into the final revision.
> > >
> > > Thank you once again for your time and insightful guidance.
> > >
> > >
> > > Best Regards,
> > >
> > > Paper 11022 Authors

---

### Official Review · Reviewer_f9WW · 2026-03-12

**Soundness:** 3
**Presentation:** 3
**Significance:** 2
**Originality:** 3
**Overall Recommendation:** 4
**Confidence:** 3

**Summary:**

This paper proposes Self-Reflective Test-Time Reinforcement Learning (SR-TTRL), a framework designed to improve pseudo-label generation in Test-Time Reinforcement Learning (TTRL). The authors argue that in complex reasoning tasks, the correct answer is often not the majority among sampled solutions, but instead belongs to a “logical minority.” To address this issue, the proposed framework replaces majority voting with a three-stage pipeline consisting of trajectory pool construction, trajectory summarization, and self-reflective verification. The resulting pseudo-labels are then used for policy optimization. Experimental results on several mathematical reasoning benchmarks show that SR-TTRL consistently outperforms majority-voting approaches, with particularly notable gains on challenging tasks such as AIME.

**Compliance With Llm Reviewing Policy:**

Affirmed.

**Key Questions For Authors:**

The authors may consider further discussing several of the question identified in weakness.

**Limitations:**

yes

**Strengths And Weaknesses:**

## Strength
1. The paper identifies an important limitation of majority voting in TTRL-based reasoning systems. The analysis suggesting that correct solutions may belong to a logical minority provides a compelling motivation for exploring alternative pseudo-label selection mechanisms. The paper clearly articulates this problem and positions the proposed method as a targeted improvement.
2. The proposed three-stage pipeline is logically organized. Each component has a clear purpose in the pipeline, and the design provides a systematic way to filter candidate reasoning trajectories and identify higher-quality pseudo-labels.
3. The experimental results cover multiple model configurations and consistently show improvements over standard TTRL baselines. The reported gains are particularly notable on challenging benchmarks such as AIME. The ablation studies also demonstrate the contribution of individual components in the pipeline, which strengthens the empirical support for the approach.

## Weakness：
1. Although the method relies heavily on self-reflection to evaluate candidate trajectories, the paper does not compare SR-TTRL with other reflection-based reasoning or verification approaches. Without such comparisons, it is difficult to assess whether the observed improvements come primarily from the proposed framework or from the general use of reflection mechanisms.

2. The proposed framework is primarily positioned as an improvement to the TTRL pipeline. As a result, it remains unclear whether the approach can generalize to broader test-time optimization or reasoning settings beyond this specific framework.

3. Several core components of the pipeline—such as trajectory clustering, summarization, and reflective evaluation—are based on existing techniques. While the combination of these steps forms a coherent framework, the paper provides limited discussion on the novel methodological contributions beyond integrating these existing operations into the TTRL setting.

4. The paper relies primarily on empirical results and does not provide theoretical analysis explaining why the self-reflection process reliably identifies logically correct trajectories. In particular, the work does not analyze whether the trajectories selected through reflection are indeed more logically consistent or closer to the ground-truth reasoning process.

5. The clustering and summarization stages are partly motivated by reducing reasoning token usage. However, the paper does not provide a detailed analysis of the overall token budget or computational cost of the proposed pipeline compared with majority-voting approaches.

6. The experiments focus primarily on math benchmarks. It would be helpful to evaluate the approach on additional domains (e.g., QA or other reasoning tasks beyond math) to better understand the generalization ability of the framework.

---

> ### Author Rebuttal · Authors · 2026-03-31
>
> ## **Q1: Comparison with General Reflection**
> To clarify, SR-TTRL is not a simple application of self-reflection; it is a structured framework (with 3 steps) specifically designed to address the statistical noise and context window limitations in reflection.
>
> To isolate the contribution of our specific framework from the general benefits of reflection, we compare SR-TTRL with Vanilla Reflection (where the model directly selects from $N$ raw trajectories without our grouping and summarization steps):
>
> || Acc (AIME24, Qwen3-8B)|
> |:--- |:---:|
> |Maj-Vote|46.7|
> |Vanilla Reflection|48.8|
> |SR-TTRL|55.8|
>
> - Our full framework outperforms Vanilla Reflection by +7.0. This gain is achieved because raw trajectories contain excessive stylistic noise and redundant tokens that often lead to "reflection failure" or context overflow in standard models.
> - Standard reflection often struggles when a wrong answer is repeated many times (consensus bias). Our "Deduplication + Candidate Pool Construction" ensures the model evaluates unique logical paths rather than being swayed by the frequency of identical incorrect responses.
> - TTRL-Specific Optimization: While general reflection methods (like Self-Refine) focus on iterative editing, SR-TTRL focuses on high-fidelity reward signal extraction from a batch of samples to guide reinforcement learning, which is a distinct objective.
>
> **Conclusion:** While "reflection" provides a baseline improvement, the structured pipeline of SR-TTRL (Summarize-then-Verify) provides a significant additional boost by making the reflection process more robust and context-efficient. We will add this comparison to our discussion on "Reflection vs. Structured Verification" in the revision.
>
> ---
> ## **Q2: Generalization and Compatibility**
>
> We demonstrate that SR-TTRL generalizes beyond the simple TTRL pipeline through its high compatibility with advanced frameworks and its effectiveness as an inference-time scaling strategy.
>
> **1. Compatibility with Advanced TTT Frameworks**
>
> SR-TTRL is orthogonal to and enhances contemporary Test-Time Training (TTT) frameworks:
>
> **(1) Co-rewarding** with multi-view supervision
>
> ||Qwen3-8B-Base on AMC|
> |:---|:---|
> |Maj-Vote|50.0|
> |Co-rewarding|53.5|
> |SR-TTRL|56.2|
> |SR-TTRL + Co-rewarding|60.3|
>
> **(2) Self-Harmony** (question paraphrasing and cross-view consistency)
>
> ||Qwen3-8B on AIME24|
> |:---|:---|
> |Maj-Vote|46.7|
> |Self-Harmony|51.3|
> |SR-TTRL|55.8|
> |SR-TTRL + Self-Harmony|56.2|
>
> **2. Generalization to Test-Time Inference Scaling**
>
> Our "Summarize-then-Verify" mechanism generalizes as a robust inference-time scaling strategy even without RL training.
>
> ||Acc (Qwen3-8B on AIME24)|
> |:---|:---|
> |Pass@64|80.0|
> |Maj-Vote|43.3|
> |SR|66.7|
>
> ---
> ## **Q3: Beyond Integration: Addressing Consensus Bias via Structured "Summarize-then-Verify" Self-Reflective Pipeline**
> Our core novelty lies in the structured "Summarize-then-Verify" pipeline specifically designed to solve the "consensus-bias" trap of majority voting in TTRL. While individual techniques exist, their integration is a non-trivial methodological contribution that enables models to identify the "Logical Minority" by filtering stylistic noise and staying within context limits, a capability that simple reflection or clustering alone cannot achieve for autonomous self-evolution.
>
> ---
> ## **Q4: Theoretical Analysis**
> Due to character constraints, we kindly invite the reviewer to refer to the detailed theoretical analysis in our response to Reviewer h9BT Q2.
>
> ---
> ## **Q5: Computational Cost Breakdown**
> Table 4 in the paper compares the time cost of TTRL and SR-TTRL. We further break down the cost of each step in SR-TTRL, please kindly refer to the detailed results in our response to Reviewer mZVG Q4, due to character constraints.
>
> ---
> ## **Q6: Additional Experiments on Other Benchmarks (science, puzzle, coding, general reasoning)**
> To demonstrate the versatility of SR-TTRL, we extend our experiments to scientific reasoning (GPQA), logic puzzles (ZebraLogic), coding (LiveCodeBench), and general QA (MMLU-Redux).
>
> The results using the Qwen3-1.7B model are summarized below:
>
> ||GPQA|ZebraLogic|LiveCodeBench-v5|MMLU-Redux|
> |:---|:---|:---|:---|:---|
> |Qwen3-1.7B|28.6|12.8|11.6|64.4|
> |+TTRL|30.5|20.3|14.2|70.8|
> |+SR-TTRL|35.7 (+5.2)|28.8 (+8.5)|20.4 (+6.2)|74.3 (+3.5)|
>
> - **Consistent Gains Across Domains:** SR-TTRL significantly outperforms standard TTRL in all tested non-math tasks, with particularly substantial improvements in logic puzzles (+8.5%) and coding (+6.2%).
> - **Universal Logic Capture:** These results prove that our "Summarize-then-Verify" pipeline is not math-specific. It successfully identifies critical logical anchors and deductive steps in scientific, algorithmic, and commonsense reasoning, providing high-fidelity reward signals for autonomous evolution.
>
> We will include these multi-domain results in the revised manuscript to further substantiate the generalization capabilities of our framework.

---

> > ### Author Rebuttal · Reviewer_f9WW · 2026-04-07
> >
> > Thank you for the detailed rebuttal.

---

### Official Review · Reviewer_h9BT · 2026-03-13

**Soundness:** 3
**Presentation:** 3
**Significance:** 2
**Originality:** 3
**Overall Recommendation:** 4
**Confidence:** 3

**Summary:**

## Summary and Overall Assessment
The paper investigates a topic on the rapidly evolving domain of Test-Time Reinforcement Learning (TTRL): the limitations of the standard "Majority Voting" mechanism in complex reasoning tasks, where correct answers frequently fall into the "logical minority." To address this, the authors propose a clever SR-TTRL framework that shifts from statistical consensus to a self-reflective logical verification process, utilizing trajectory pooling, summarization, and cross-comparison. The authors have identified a highly intuitive and practical bottleneck in current reasoning scaling laws. I truly appreciate it. However, to firmly establish its contribution at a venue like this, the paper needs to address a few missing comparisons—specifically regarding contemporary baselines, theoretical grounding, and a deeper dive into its failure modes. I am currently leaning towards a weak reject, but I am very open to raising my score if these aspects can be clarified during the rebuttal.

**Compliance With Llm Reviewing Policy:**

Affirmed.

**Final Justification:**

Initially, the manuscript's method is intuitive with positive improvement compared with traditional TTRL. While I think the main weakness is lacking strong baseline comparison and theoretical explanation. In the rebuttal, author adds one dataset experiment compared with strong baseline and simple theoretical explanation, these new evidences makes the whole method more reasonable. Hence, I adjust my score from 3 to 4. Good luck.

**Key Questions For Authors:**

## Questions:
1. Can the authors provide empirical comparisons against more recent, advanced TTRL baselines(like [Co-rewarding](https://openreview.net/pdf?id=fDk95XPsCU) , [Self-Harmony](https://openreview.net/pdf?id=ZzG6oJ5ehI) and so on} that do not rely on simple majority voting?
2. Although Table 4 has makes a good speed analysis, since the paper states SR-TTRL works well at $N=8$ compared to baseline $N=64$, can author compare the speed between $N=8$ of SR-TTRL and $N=64$ of TTRL(at same level accuracy)?
3. Could you provide a breakdown of the errors when pseudo-labeling fails? Specifically, what is the main bottleneck preventing the method from reaching the $Pass@N$ potential limit? Which module (e.g., faulty summarization, or the final self-reflection step hallucinating a logical link) is the primary culprit for these failures?
4. Could you give me a theoretical analysis why SR-TTRL can overcome(improve) the failure modes of majority voting?

**Limitations:**

Yes

**Strengths And Weaknesses:**

## Strengths
- Strong and Insightful Motivation: The analysis of the "logical minority" and the scaling divergence between a model's potential ($Pass@N$) and its consensus capacity ($Maj@N$) is excellent. It clearly articulates why majority voting hits a ceiling.
- Elegant Engineering: The pipeline design is highly practical. Using summarization to condense trajectories before feeding them into a cross-verification prompt is a smart way to bypass the context window limits and attention degradation typically seen in long reasoning tasks.

## Weaknesses
- Need of Contemporary Baselines: The limitations of majority voting in TTRL have been widely recognized in recent literature leading to the development of alternative verification methods like Co-Reward, or Self-Harmony. Comparing SR-TTRL solely against a standard majority-voting baseline is insufficient to prove state-of-the-art effectiveness.
- Lack of Theoretical Analysis: While the empirical results are very encouraging, the paper is primarily driven by heuristics and empirical observations. Providing some theoretical intuition or formal analysis detailing why and under what data distributions the self-reflection mechanism outperforms majority voting would significantly elevate the paper's scientific rigor.
- Need for Concrete Examples: The paper describes the modules (Candidate Trajectory Pool Construction, Trajectory Summarization, Self-Reflective Verification) abstractly. Adding a concrete, end-to-end qualitative example in the main text (or appendix) showing exactly what the inputs and outputs look like at each of these steps would greatly enhance readability and reproducibility.

---

> ### Author Rebuttal · Authors · 2026-03-31
>
> ## **Q1: Comparison with Co-rewarding and Self-Harmony**
> ### **(1) SR-TTRL vs. Co-rewarding**
> **Method:** Co-rewarding uses multi-view supervision, but it still uses majority voting pseudo-labeling in each view. SR-TTRL is a fundamental improvement over majority-voting, which is ***orthogonal*** to Co-rewarding. While Co-rewarding provides diverse views, SR-TTRL ensures higher-fidelity labels within each view.
>
> **Experiment:** We follow the same test-time training settings in Co-rewarding for fair comparison. SR-TTRL outperforms Co-rewarding by +2.7, showing its superiority. Combining the two yields 60.3, proving that our method is highly compatible with and enhances advanced frameworks.
>
> ||Qwen3-8B-Base on AMC|
> |:---|:---|
> |Maj-Vote|50.0|
> |Co-rewarding|53.5|
> |SR-TTRL|56.2|
> |SR-TTRL + Co-rewarding|60.3|
>
> ### **(2) SR-TTRL vs. Self-Harmony**
>
> **Method:** Self-Harmony leverages multi-view consistency (original + paraphrased question) to filter spurious answers via harmonic mean aggregation. However, it still fundamentally relies on statistical frequency across views, which cannot resolve the "logical minority" problem where correct answers appear infrequently. Our SR-TTRL represents a paradigm shift from frequency-driven to quality-driven pseudo-labeling, using self-reflective verification to identify logically sound trajectories regardless of their frequency. These two approaches are ***complementary***: Self-Harmony serves as an efficient coarse filter to remove obviously wrong candidates, while SR-TTRL acts as a precise fine-grained validator to select the most rigorous reasoning path. ***Combined approach***: Self-Harmony filtering applied between Step 1 and Step 2 of SR-TTRL to prune low-confidence candidates before self-reflective verification.
>
> **Experiment:** As table below, SR-TTRL consistently outperforms Self-Harmony. Moreover, combining both methods achieves better performance than either alone, confirming their complementary nature.
>
> ||Qwen3-8B on AIME24|
> |:---|:---|
> |Maj-Vote|46.7|
> |Self-Harmony|51.3|
> |SR-TTRL|55.8|
> |SR-TTRL + Self-Harmony|56.2|
> ---
> ## **Q2: Theoretical Analysis**
> We provide a formal theoretical analysis to demonstrate why our Self-Reflective (SR) mechanism overcomes the limitations of Majority Voting (MV).
>
> **(1) Data Distribution and MV Failure:**
>
> In complex reasoning, models often encounter "Logical Traps" (Skewed Distributions), where a systematic error $a_{trap}$ is more frequent than the ground-truth $a^{\*}$. Formally:
> $\exists a_{trap} \in \mathcal{A} \setminus \{a^{\*}\} \ \ \text{ s.t. } \. p(a_{trap}) > p(a^{\*}) > 0$
> Under this distribution, ***as sample size $N \to \infty$***, the empirical frequency of $a_{trap}$ dominates. Thus, ***the probability of MV selecting the correct answer approaches zero***: $\lim_{N \to \infty} P(\hat{a}_{MV} = a^{\*}) = 0$. This explains the "Scalability Paradox" where increasing $N$ fails to improve pseudo-label quality.
>
> **(2) Why SR Improves MV's Failure:** SR disentangles selection from frequency through grouping and logical verification.
> - **Deduplication:** By retaining one representative per answer, the correct solution enters the verification stage with probability $P(a^{\*} \in \mathcal{P}) = \text{Pass}@N = 1 - (1 - p(a^{\*}))^N$.
> - **Logical Consensus:** Let $\gamma$ be the model’s Verification Competence (the probability of correctly identifying sound logic over fallacies). The success probability is:
> $P(\hat{a}_{SR} = a^{\*}) \approx \gamma \cdot [1 - (1 - p(a^{\*}))^N]$
> Since verifying a logic skeleton is fundamentally easier than generating a full trajectory ($\gamma \gg p(a^{\*})$), ***SR’s accuracy increases monotonically with $N$***. This proves SR breaks the statistical ceiling of frequency-based methods.
>
> ---
> ## **Q3:Qualitative Examples of SR-TTRL Modules**
> Due to the character limit for Rebuttal, we will provide end-to-end qualitative examples for each module in the revised Appendix and are ready to share them during the discussion phase if requested.
>
> ---
> ## **Q4: Speed Comparison between SR-TTRL (N=8) and TTRL (N=64)**
> SR-TTRL is 2.1x faster than TTRL to reach identical performance.
>
> ||N| Acc|Time per step|
> |:---|:---|:---|:---|
> |TTRL|64|46.7|278s|
> |SR-TTRL|8|46.8|132s|
>
> ---
> ## **Q5: Bottleneck Analysis for the Accuracy of Self-Reflective (SR) Pseudo-labeling**
> The primary bottleneck is "Self-Reflective Verification", not "Summarization".
> - We use GPT-5.2 to evaluate the faithfulness of summarized skeletons. Around 95.7% error steps from the raw trajectories are retained (highly faithful). Thus, the gap to Pass@64 is almost due to "Verification".
> - The summarization step is indispensable, despite minor information loss. From Table below, removing this step causes a -6.7 drop. Summarization remains a critical "noise-filter" that enables the verifier to operate closer to its theoretical maximum.
>
>
> ||Pseudo-Label Acc (Qwen3-8B on AIME24)|
> |:---|:---|
> |Pass@64|80.0|
> |Maj-Voting|43.3|
> |SR|66.7|
> |SR w/o Sum|60.0|

---

> > ### Author Rebuttal · Reviewer_h9BT · 2026-04-04
> >
> > The author directly uses new experiments result and discussion resolving my issue. Hope author can add all these modification into revision version(Comparison with Co-rewarding and Self-Harmony in all dataset shown in main context, theoretic explanation). I will raise my score assuming author will add it. Good luck.

---

> > > ### Author Response · Authors · 2026-04-04
> > >
> > > Dear Reviewer h9BT,
> > >
> > >
> > >
> > > We sincerely thank you for the positive evaluation and for raising the score. We are pleased that your concerns have been successfully resolved.
> > >
> > > We firmly commit to incorporating all discussed modifications into the final revision, including the comprehensive performance comparisons, theoretical analysis, qualitative examples, detailed speed analysis, and pseudo-labeling bottleneck analysis.
> > >
> > > Thank you once again for your time and valuable suggestions.
> > >
> > >
> > >
> > > Best Regards,
> > >
> > > Paper 11022 Authors

---

### Decision · Program_Chairs · 2026-04-30

**Decision:**

Accept (regular)

**Comment:**

This paper proposes SR-TTRL, replacing majority voting with a self-reflective verification mechanism to better identify “logical minority” solutions in test-time reinforcement learning. Reviewers generally find the motivation clear and insightful, and the empirical improvements over standard TTRL are consistent across multiple benchmarks.

Several concerns were raised, including whether the gains mainly come from generic self-reflection, limited novelty as the method combines existing components, and questions about evaluation scope and generalization beyond math reasoning. There were also concerns regarding reliance on the model’s own verification ability and the overall computational overhead.

The rebuttal addressed most of these points with additional experiments and clarifications (e.g., thinking-mode results, external verifiers, and broader evaluations), and reviewers indicated their concerns were largely resolved. Overall, while some limitations remain, I lean toward weak accept.